# Simulation of Sentinel-2 Bottom of Atmosphere Reflectance Using Shadow Parameters on a Deciduous Forest in Thailand

**Takumi Fujiwara \* and Wataru Takeuchi**

Institute of Industrial Science, The University of Tokyo, 4-6-1 Komaba, Meguro-ku, Tokyo 153-8505, Japan; wataru@iis.u-tokyo.ac.jp

\* Correspondence: fujiwara-takumi930@g.ecc.u-tokyo.ac.jp

**Abstract:** The shadow fraction of the canopy is an important factor in Bidirectional Reflectance Distribution Function (BRDF) and in estimating physical quantities, such as tree height and biomass. Shadows are used as a shielding ratio for direct solar irradiance, but, at shorter wavelengths, the amount of diffuse solar irradiance is greater, so the shielding ratio cannot be ignored. The shielding ratio of direct and diffuse solar irradiance is called Cast Shadow (CS) and Self Cast Shadow (SCS), respectively; however, it has been pointed out that the effect of these shadows is greater at higher resolutions, such as Sentinel-2. In addition, the Bottom Of Atmosphere (BOA) reflectance is greatly affected by shadows, because it is corrected for atmospheric effects. Therefore, the main purpose of this study is to investigate the spatial variability of CS and SCS and simulate the Sentinel-2 BOA reflectance with these shadows. The target forest was a greenness season of a deciduous broadleaf forest in Thailand. First, we obtained a point cloud of the forest by Structure from Motion while using the Unmanned Aerial Vehicle. Next, we created a voxel model with CS and SCS as attributes. CS was calculated as the percentage of area where the plane that is assumed per voxel is shielded from direct solar irradiance by other voxels. SCS was calculated as the percentage of area where the hemispheric radiant environment is shielded by other voxels. Subsequently, using solar irradiance and leaf spectral reflectance data, the reflectance of each band of Sentinel-2 was simulated. Nine leaves were used to investigate the effect of leaf species on the simulation. The reflectance acquired by Sentinel-2 is not at the leaf level; however, we used this spectral reflectance data because the reflectance was simulated at the same spatial resolution as the voxel size. Voxel sizes of 20 cm, 50 cm, 100 cm, and 200 cm were used. Our result showed that (1) the spatial variability of SCS was smaller than that of CS when the sun position is fixed and the view zenith angle is changed. SCS was mostly 0.12 at different zenith angles, while the CS had a maximum value of 0.45 and a minimum value of 0.15. (2) The accuracy of the simulations was evaluated using the Root Mean Square Error (RMSE). The best RMSE is $0.020 \pm 0.015$ and the worst one is $0.084 \pm 0.044$. It was found that the error is larger in short wavelength infrared bands. (3) In this forest, the relative reflectance changed only about 1.2 times as much, as the voxel size was increased from 20 cm to 200 cm. In this study, we have simulated a single Sentinel-2 image. In the future, we will simulate multi-temporal images in order to investigate the effects of phenology and shadow changes on the reflectance that was observed by optical sensors.

**Keywords:** Sun-Target-Sensor-Geometry; shadow; point cloud; Unmanned Aerial Vehicle; spatial variation

## 1. Introduction

The satellite data that were obtained from optical sensors are influenced by the shadows caused by Sun-Target-Sensor-Geometry. Ignoring the effect of shadows on radiance results in incorrect

estimates of object properties, such as reflections [1,2]. In particular, high-resolution satellite images have been used in recent years, but, as scenes become more complex, the effects of shadows also increase exponentially [3]. Therefore, shadow correction is still an important topic for urban areas and mountainous forests [4]. On the other hand, some studies have examined the relationship between the proportion of shadows in pixels and vegetation type [5] or sky view factor in urban areas [6]. Ono et al. [7] have proposed the Shadow Index (SI) while using the proportion of radiation that was attenuated by shadows and calculated correlations with tree height for each major vegetation type using Moderate Resolution Imaging Spectroradiometer (MODIS) images. The results showed that, for most vegetation, the correlation coefficients between SI and tree height were higher than those of the Normalized Difference Vegetation Index (NDVI). Some studies have used the Bi-directional Reflectance Distribution Function (BRDF) product of MODIS in order to estimate tree height [8,9]. The BRDF product of Sentinel-2 and Landsat recently has been developed by combining with that of MODIS [10–12]; however, BRDF correction has not yet been performed while only using those resolution images, because a minimum of seven valid images is needed from the 16 days of reflectance data from multiple directions [13]. On the other hand, because the shadow proportion can be calculated from a single image, it is possible to apply in various of satellite images. Particularly, in forests, shadows are related to complex three-dimensional structures, so it is expected that various parameters, such as foliage clumping or light use efficiency [14] other than tree height, can be extracted. Shadow proportion is also an important factor in ecosystem functions, such as photosynthesis and evapotranspiration [15]. Therefore, understanding the relationship between shadow and satellite sensor data is necessary in order to correct shadows in pixels or to estimate the physical quantity of the forest using that one.

Various simulators have already been developed to mimic the interaction between the radiance and forests (e.g., The Discrete Anisotropic Radiative Transfer Model (DART) [16], Forest Light Environmental Simulator (FLiES) [17], and Radiosity-Graphics combined method (RGM) [18]. These simulators use vegetation parameters, such as leaf area index and leaf area density, into a virtual forest model and calculate radiance by ray tracing. However, to our knowledge, the relationship between shadow and radiance or reflectance has not been investigated, because shadow is not used as a parameter.

Representing the three-dimensional structure of the forest is necessary in order to simulate shadows. For this purpose, point clouds that can be acquired by elemental techniques, such as Light Detection And Ranging (LiDAR) and Structure from Motion (SfM), are suitable. Various studies have investigated the spatial distribution of structural parameters while using these elemental techniques [19,20]. However, the point clouds acquired by LiDAR and SfM need to be converted to other data structures. Mesh and voxel models are often used as alternative data structures to 3D models, because point cloud data are spatially heterogeneous and the amount of data is huge. In the case of mesh models, the object must be covered with triangles, and any missing data must be manually completed [21]. In the case of voxel models, on the other hand, it is not necessary to do so because it is a 3D raster domain, a discrete 3D space containing elements [22]. Using a voxel model, the method to estimate various vegetation parameters such as leaf area [23], woody material volume [24], leaf inclination angle [25], and Sun-Induced chlorophyll fluorescence [26] have already been developed.

Wang et al. [27] have recently developed a voxel-based sunlit and shade component approach while using Aerial Laser Scanning data (ALS) to investigate the spatial variability of forest sunlit and shadow components. Usually, shadow is used as a shielding ratio for direct solar irradiance, but, at short wavelengths, the amount of diffuse solar irradiance increases, so its shielding ratio cannot be ignored. In addition, the Bottom Of atmosphere (BOA) reflectance is greatly affected by shadows, because it is corrected for atmospheric effects. Arévalo et al. [28] have categorized shadows into Cast Shadow (CS), which is caused by projecting a shadow onto another object, and Self Cast Shadow (SCS), which is caused by self-projection. Therefore, in this study, CS and SCS were regarded as the shielding

ratio of direct and diffuse solar irradiance, respectively. In addition, CS and SCS were reproduced with a spatial resolution of several tens of centimeters by acquiring a point cloud by SfM.

The objectives of this study were to (1) investigate the spatial variations of CS and SCS; (2) compare of satellite-observed and simulated reflectance; and, (3) investigate the effect of voxel size on simulation result. The target satellite image was Sentinel-2 and its BOA reflectance was used.

## 2. Methodology

### 2.1. Flow Chart of This Study

Figure 1 shows the flowchart of this study. Firstly, the point cloud of the target forest was obtained by Unmanned Aerial Vehicle (UAV)-SfM. The reason for using the UAV-SfM is that it can observe a large area of forest in a short time. In addition, it was assumed that the effect of the shadow on the tree canopy would have a large effect on the reflectance. Secondly, the point cloud data were registered to voxel coordinates. The attributes of the voxel model were CS and SCS. CS and SCS means shielding ratio of direct and diffuse solar irradiance, respectively. The methods for creating the voxel model and computing the CS and SCS were described in Section 2.4. Thirdly, after deciding the viewing position, the voxel model was converted to image. Fourthly, the Sentinel-2 BOA reflectance was simulated while using solar irradiance, leaf reflectance, and the Spectral Response Function (SRF) of that sensor. In this study, all of the voxels were assumed as leaf voxel. Section 2.5 describes the reflectance simulation method using CS and SCS. The Sentinel-2 Top Of Atmosphere (TOA) reflectance product used was converted to BOA reflectance while using the Sentinel Application Platform (SNAP) tool.

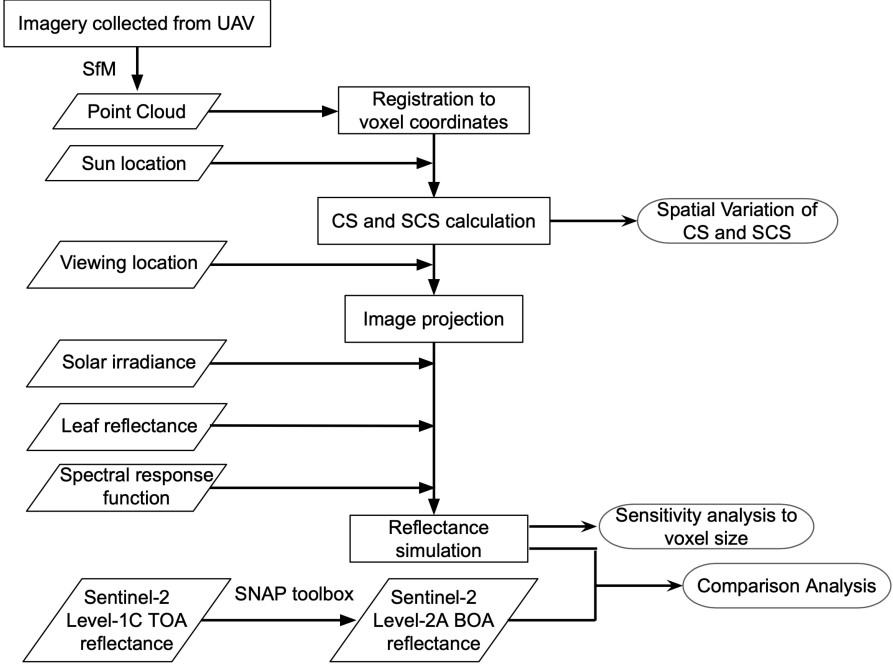

**Figure 1.** Flowchart of this study. Voxel model was created based on the point cloud and Cast Shadow (CS) and Self Cast Shadow (SCS) were computed as attributes data. The spatial variation of these attributes was calculated with limited view zenith angle and fixed view azimuth angle. The simulated reflectance with shadow parameters was compared with Sentinel-2 BOA reflectance. Finally, the effect of voxel size on simulation result was investigated.

### 2.2. Study Site

Our study forest is located in the Suranaree University of Technology in Nakhon Ratchasima (14.88° N, 102.01° E), Thailand (Figure 2). The site is flat and there is no mountain around the forest. The target extent is 170 m × 130 m and the elevation is about 230 m. The stand in the forest is deciduous

dipterocarp. Some studies have been done on tropical forests [29,30], but, to our knowledge, no UAV observations or 3D modeling have been done on deciduous dipterocarp in Thailand. Figure 2a shows the location of the study site and the overall of the study forest. The orthophoto was obtained by UAV observation. Figure 2b shows a photograph that was taken by near the target forest. From this photo, it can be seen that the inside of the forest is also densely with leaves.

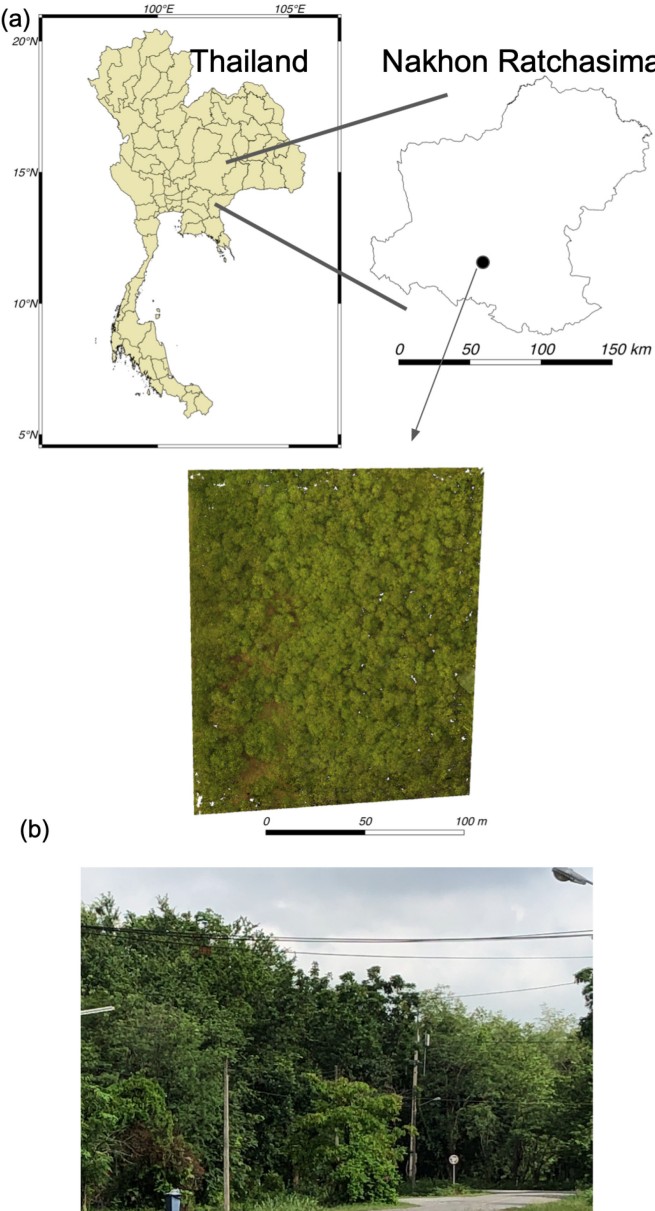

**Figure 2.** (**a**) The location of the study area and overall the study forest. The area is located in the Suranaree University of Technology in Nakhon Ratchasima, Thailand. The site is dense forest and soil is exposed on some area. The main stand is deciduous dipterocarp forest. (**b**) A photograph taken by near target forest.

*2.3. Data Collection*

2.3.1. UAV Observation

UAV observation was done on 19 September 2019 in the target forest. Weather condition was cloudy. The UAV used is PHANTOM 4 products by DJI equipped with a Global Positioning Satellite (GPS) receiver. The height from ground was about 50 m when took a photograph using the UAV.

Totally, 816 images which was 4000 × 3000 pixels were taken with slant nadir view (80°) and average ground sampling distance was 2.26 cm. The software used for generating point cloud was Pix4D mapper Pro version 4.1.22 (Pix4D, Lausanne, Switzerland). Geometric correction was done with referencing GPS log data without ground control point, so that geometric accuracy was not calculated. Instead of calculation that, the absolute geolocation variance [31] was evaluated.

2.3.2. Sentinel-2 Image and Its Spectral Response Function

Sentinel-2 B, level 1 C (ID = L1C_T47PRS_A013386_20190929T03438) acquired on 29 September 2019, was downloaded from Copernicus Open Access Hub (https://scihub.copernicus.eu/). Because the interval between the date of this Sentinel-2 image was acquired and the date of UAV observation is 10 days, the condition of the forest is probably the same. In this study, the bands are 2, 3, 4, 5, 6, 7, 8, 8A, 11, and 12. The central wavelengths of each band are 490 nm, 560 nm, 665 nm, 705 nm, 740 nm, 783 nm, 842 nm, 865 nm, 1610 nm, and 2190 nm, respectively. Thus, VISible (VIS), Near InfraRed (NIR), and Short Wavelength InfraRed (SWIR) were the simulated target wavelength. The imagery was atmospherically corrected while using the Sen2Cor tool, which is available in the SNAP toolbox. Sen2Cor reads parameters in the form of Look Up Tables (LUTs) shown in Table 1 and performs atmospheric corrections; the LUTs are generated via libRadtran, a library for calculating solar and thermal radiation in the Earth's atmosphere [32]. SRF was obtained from https://earth.esa.int/web/sentinel/user-guides/sentinel-2-msi/document-library/-/asset_publisher/Wk0TKajiISaR/content/sentinel-2a-spectral-responses (version 3.0 released 19 December 2017). The interval of the wavelength was one nm.

**Table 1.** Parameter space for atmospheric correction using Sen2Cor [32]. Some of the parameters were determined based on the image meta data.

| Parameter | Range | Increment/Grid Points |
|---|---|---|
| Solar zenith angle | 0–70° | 10° |
| Sensor view angle | 0–10° | 10° |
| Relative azimuth angle | 0–180° | 30° (180° = backscatter) |
| Ground elevation | 0–2.5 km | 0.5 km |
| Visibility | 5–120 km | 5, 7, 10, 15, 23, 40, 80, 120 km |
| Water vapor, summer | 0.4–5.5 cm | 0.4, 1.0, 2.0, 2.9, 4.0, 5.0 cm |
| Water vapor, winter | 0.2–1.5 cm | 0.2, 0.4, 0.8, 1.1 cm |

2.3.3. Spectral Reflectance of the Leaves

Hyperspectral images are often used to acquire spectral reflectance data for an area of interest [27]. However, the SRF of the sensor used to acquire the images is different from the SRF of Sentinel-2, which can cause errors. Therefore, in this study, we selected spectral reflectance data for the simulation from the ECOSTRESS Spectral Library (version 1.0) (https://speclib.jpl.nasa.gov). However, no spectral reflectance data were available for leaves of the same tree species as the forest in the target area. Therefore, leaves from three genera with deciduous species, (*Quercus*, *Fagus*, and *Betula*), were selected. To investigate the variation of the simulation results with different leaf species, nine leaves types were selected, as shown in Figure 3. The wavelength interval is one nm.

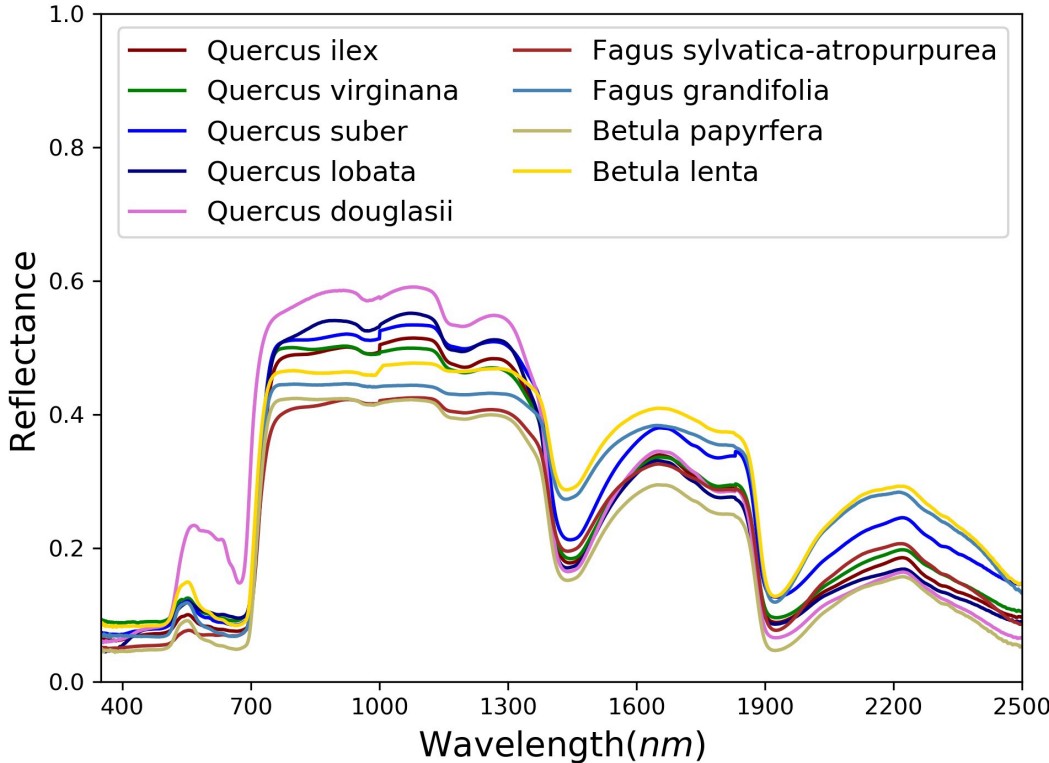

**Figure 3.** Spectral reflectance data selected from the ECOSTRESS Spectral Library used for the simulations. The main leaf genera are Quercus, Fagus, and Betula. In total, nine species of leaves were used. The wavelength interval is one nm.

*2.4. Voxel Model Creation*

The point clouds that were obtained by UAV-SfM were registered in voxel coordinates using the following equations. In this study, the voxel size is determined, so that. at least one point is within the voxel, taking into account the average density of the obtained point cloud. After registration, CS and SCS were calculated while using the average and center coordinates of the point clouds in the same voxel array. Reducing the voxel size improves the accuracy of the CS and SCS calculations, but the number of point clouds in the voxel decreases.

$$i = floor(\frac{X - X_{min}}{\Delta i}) \tag{1}$$

$$j = floor(\frac{Y - Y_{min}}{\Delta j}) \tag{2}$$

$$k = floor(\frac{Z - Z_{min}}{\Delta k}) \tag{3}$$

where $i$, $j$, $k$ are the coordinates within the voxel array, $X$, $Y$, $Z$ are coordinates of the point cloud, $X_{min}$, $Y_{min}$, $Z_{min}$ are the minimum value of $X$, $Y$ and $Z$, respectively, and $\Delta i$, $\Delta j$, $\Delta k$ are the voxel size.

2.4.1. Voxel Based Computation of Self Cast Shadow

SCS represents the shielding ratio of diffuse solar irradiance as it enters the voxel. Thus, it is the same concept as the sky view factor, which is the ratio of radiation that is received by a planar surface from the sky to that received from the entire hemispheric radiating environment [33]. This indicator has been used for evaluating the light environment of a forest floor [34] or microclimate of a city [35].

In this study, the method that was developed by Fujiwara et al. [36] was used to calculate SCS per voxel. The method projects shielding situation of a diffuse horizontal irradiance (Figure 4a) onto an

image (Figure 4b). The image's plane (Figure 4b) was polar coordinates, which the distance from the center and declination correspond to altitude angle and azimuth angle of direction vector (Figure 4a). The circle on the image represents an altitude angle of zero degrees. After a cylinder, which, in the same direction as direction vector, was generated, whether exist voxel between target voxel and sky was discriminated using the Equation (4). If a voxel exists, the attribute of the pixel corresponding with that direction is one. In other words, SCS means the percentage of pixels with an attribute of one inside that circle. The radius of the cylinder was the same as the inscribed sphere radius of the voxel.

$$|(\vec{p} - \vec{p_0}) \times \vec{v}| \leq r \tag{4}$$

where $\vec{p}$ is position vector of average coordinate of each voxel, $\vec{p_0}$ is position vector of target voxel, $\vec{v}$ is direction vector from the viewpoint and $r$ is radius of cylinder.

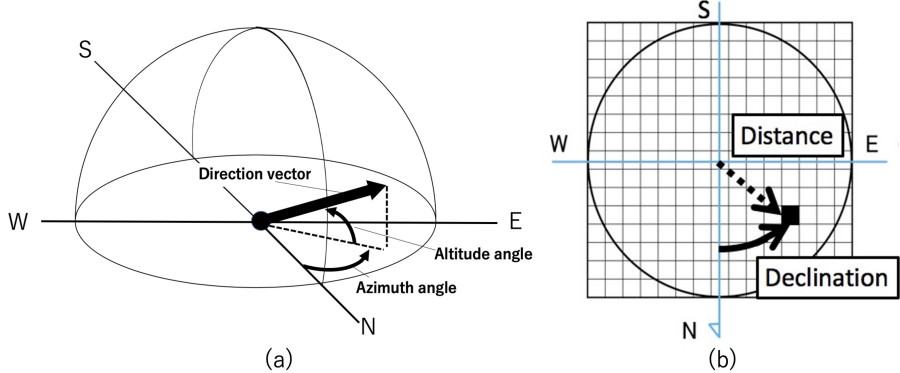

(a)               (b)

**Figure 4.** Conceptual diagram of SCS computation. SCS indicates the proportion of diffuse solar irradiance that is shielded by other voxels. (**a**) Determine a directional vector from the viewpoint voxel and generate a cylinder in that direction. (**b**) According to the discriminant shown in Equation (4), it is determined whether that direction is shielded by other voxels or not. The result was projected onto the image. The pixel value was zero or one, which means be unshielded or shielded, respectively. The distance from the center of the image and the declination angle correspond to the altitude angle and horizontal angle in (**a**).

### 2.4.2. Voxel Based Computation of Cast Shadow

CS indicates the shielding ratio of direct solar irradiance enters voxel. In this study, we used the method that was proposed by Fujiwara and Takagi. [37] in order to calculate CS using a voxel model. The method is as follows. Firstly, a plane, which, parallel to the XY plane, was assumed at the average Z coordinate position of the voxel. Thus, the direction of the normal vector of the plane is vertical upward. Secondly, the plane was equally divided into four, and a line is generated from each plane toward the sun. Thirdly, a point of the intersection of the line with another voxel's plane was calculated. If the point was inside the voxel, the line was intersecting. Finally, among the four straight lines, the shadow ratio was determined by the number of lines intersecting the plane of other voxels. Thus, the CS value is one of 0.0, 0.25, 0.5, 0.75, or 1.0. For example, in case of Figure 5, the CS value is 0.75.

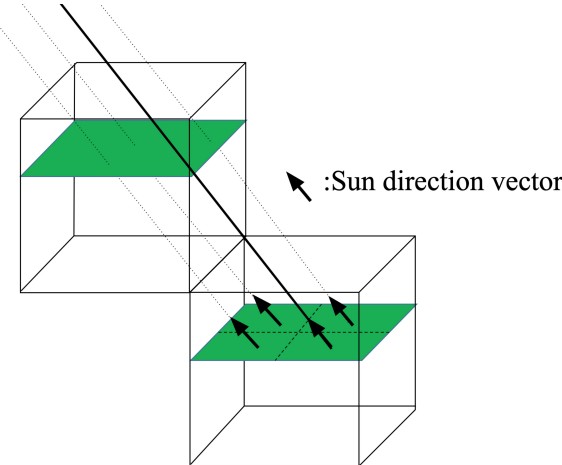

**Figure 5.** Conceptual diagram of CS computation. CS indicates the fraction of direct solar irradiance that is shielded by other voxels. The plane assumed in the voxel is divided into four equal parts, and a line is generated from each plane towards the sun. The percentage of lines shielded by other voxels is the value of CS. In this method, CS is a discrete value, with a one of 0.00, 0.25, 0.50, 0.75, or 1.00. Dotted lines indicate intersections with other voxels, and bold lines indicate no intersections with one. Thus, the CS value of this voxel is 0.75.

*2.5. Reflectance Simulation Using a Shadow Parameters*

The voxel model was projected onto the image with a pixel size same with voxel size, after calculating the nadir view because Sentinel-2 level 1 C product is ortho-corrected. Reflectance was simulated following equations. Firstly, radiance that was reflected from target forest was derived from Equation (5). Voxel was assumed Lambertian surface. Direct and diffuse horizontal irradiance input to Equation (5) was predicted while using the Simple Model of the Atmospheric Radiative Transfer of Sunshine (SMARTS) code [38], version 2.9.5. SMARTS simulate these irradiances on the ground by input several atmospheric parameters shown in Table 2. The wavelength range that can be simulated is 280 to 4000 nm. Secondly, the radiance that was observed by satellite was derived from Equation (6), which considers the spectral response function. Thirdly, reflectance was derived from Equation (7). Finally, pixel size was downsampled to each band's resolution using the average value. The spatial resolution of bands 2, 3, 4, and 8 are 10 m, and other bands are 20 m.

$$I_{tar}(\lambda) = \{I_{dir}(\lambda)(1 - CS) + I_{dif}(\lambda)(1 - SCS)\}\frac{\rho(\lambda)}{\pi} \tag{5}$$

$$I_{ob} = \frac{\int I_{tar}(\lambda)SRF(\lambda)d\lambda}{\int SRF(\lambda)d\lambda} \tag{6}$$

$$r = \frac{\pi I_{ob}}{I_{dir} + I_{dif}} \tag{7}$$

where $\lambda$ is the wavelength, $I_{tar}(\lambda)$ is the radiance reflected from target forest, $I_{dir}(\lambda)$ is direct horizontal irradiance (W/m$^2$/μm), CS is shielding ratio of $I_{dir}(\lambda)$, $I_{dif}(\lambda)$ is diffuse horizontal irradiance (W/m$^2$/μm), SCS is shielding ratio of $I_{dif}(\lambda)$, $\rho(\lambda)$ is spectral reflectance of the target, $I_{ob}$ is radiance observed by sensor (W/m$^2$/μm), R($\lambda$) is spectral reflectance of the leaves, SRF ($\lambda$) is spectral response function, and *r* is reflectance.

**Table 2.** Input parameters of an atmospheric layer for run SMARTS, version 2.9.5. The output parameters are direct and diffuse horizontal irradiance. Simulable spectral range is 280 to 4000 nm, and the effective interval is 0.5 nm (280–400 nm), 1 nm (400–1700 nm), and 5 nm (1705–4000 nm).

| Input Condition | Values | Remarks |
|---|---|---|
| Station Pressure | 1013.25 mb | Basic value |
| Altitude | 0 km | Sea level |
| Reference Atmosphere | Tropical | Water Vapor, Ozone, Gas absorption and pollution |
| Carbon Dioxide | 370ppmv | Basic value |
| Aerosol Model | Rural | Shettle and Fenn [39] |
| Atmospheric Turbidity | Aerosol Optical Depth at 500 nm = 0.084 | Basic value |
| Albedo | 1.0 | Totally reflect |
| Spectral Range | 400 to 4000 nm | |
| Solar Constant | 1366.1 W/m2 | Basic value |
| Solar geometry | Zenith angle 34.2° Azimuth angle 134.0° | |

## 3. Results and Discussion

### 3.1. Point Cloud Profile and Voxel Size Determination

In total, 3,153,374 points were generated by Pix4D mapper and its average density was about 285.64 points/m$^3$. As the geolocation errors, mean, sigma, and Root Mean Square Error (RMSE) were calculated. The mean values of X, Y, and Z were 0.00 m, respectively. Sigma and RMSE values of X, Y, and Z were 2.11, 2.18, and 8.28 m, respectively. The geolocation error means the difference between the initial and computed image positions. The image geolocation errors do not correspond to the accuracy of the observed point cloud.

As mentioned in Section 2.4, there must be at least one point in the voxel. In the case of the average density of the point cloud acquired, for voxel sizes of 10 and 20 cm, the average number of points was 0.28 and 2.2, respectively. Therefore, the voxel size was set to 20 cm.

### 3.2. Spatial Variation of Cast and Self Cast Shadow

Figure 6 shows the voxel model (colors are based on the RGB of the point cloud acquired by SfM) and the spatial distribution of CS and SCS calculated as attributes. The voxel size is 20 cm. The sun position parameter for computing CS was determined based on the Sentinel-2 image metadata file (MTD_TL.xml) used. The zenith angle and azimuth angle of the Sun are 34.2° and 134.0°, respectively. Figure 6 also shows the positional relationship between the sun and voxel model. The SCS only varies with roughness and density. In other words, the value of SCS does not change as the sun position changes. The CS values have five levels: 0.00, 0.25, 0.50, 0.75, and 1.00. The SCS values are a real number between 0.00 and 1.00. From Figure 6, it can be visually seen that most of the CS value is 0.00, followed by 1.00, and most of the CS value is about 0.5.

Figure 7 shows the spatial variations of the CS and SCS shown in Figure 6 when observed at limited view zenith angles from −30° to 30° under the fixed view azimuth angle 290°. Spatial variation means the change in the mean value of the CS and SCS of the voxels visible from the sensor position at each zenith angle. The zenith angle was varied every 10°. The result shows that the trends in the spatial variability of CS and SCS are the same, but the amount of variability is different. Between −10° and 10° zenith angle, SCS remains almost the same, at 0.12, while the CS increased from 0.17 to 0.23. The mean value of SCS was the lowest at 0° zenith angle and highest at −20° and 20°. They are 0.12 and 0.15, respectively. On the other hand, the mean value of CS showed the lowest value of 0.15 at a zenith angle of −30° and the highest value of 0.45 at 20°. In addition, the mean value of CS is about twice as large at 20° than at −20° at the zenith angle. The smaller variability of the SCS when compared to the CS may be due to the higher forest density and smaller roughness.

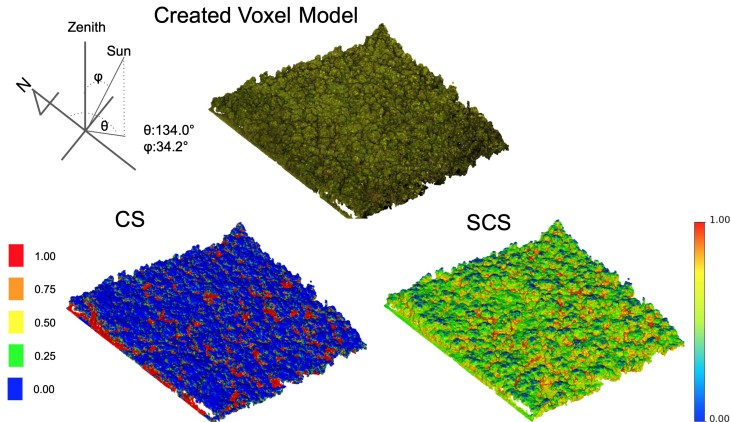

**Figure 6.** Created voxel model of the study forest and spatial distribution of CS and SCS. CS was calculated based on the position of the sun when Sentinel-2 observed. The azimuth and zenith angle are 134.0° and 34.2°, respectively. Spatial distribution of CS, the most frequent value is 1.0, followed by 0.0, which can be visually seen. On the other hand, SCS was around 0.5 for most voxels.

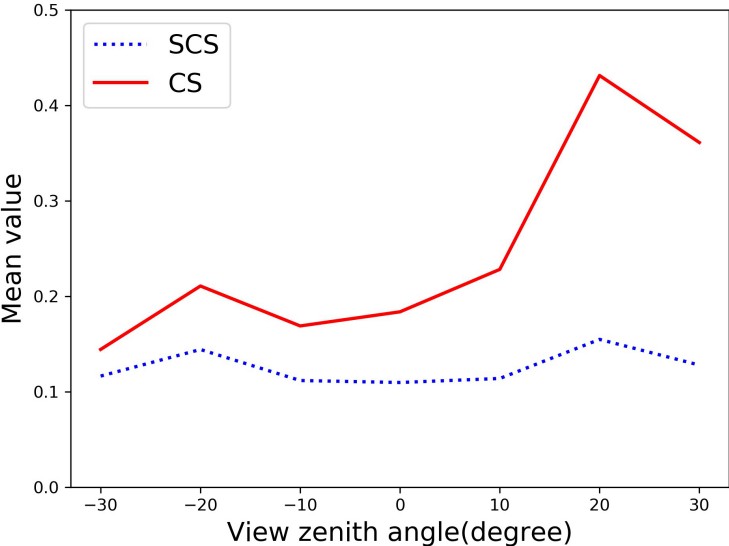

**Figure 7.** The spatial variation of SCS (blue dotted line) and CS (red line) values with the view zenith angles ranging from −30° to 30° and azimuth angle of 290°. The x-axis is the view zenith angle and the y-axis are the average of CS and SCS. As compared to CS, the spatial variability of SCS was smaller.

### 3.3. Result of Simulated Sentinel-2 BOA Reflectance

The voxel model shown in Figure 6 was used to simulate the Sentinel-2 image. As an example, Figure 8 shows simulated images of bands 3, 8A, and 11 while using the *Quercus ilex* spectral reflectance. The spatial resolution of band 3 is 10 m, while that one of band 8A and 11 is 20 m. The range of reflectance is 0.02 for band 3, while that one is 0.1 for band 8A and 11. The band 3 and 11 images are brighter than the Sentinel-2 image, which indicated that the reflectance of the simulated image is higher than the Sentinel-2 one. The band 8A image was similar in brightness to Sentinel-2 image. A comparison of images simulating band 8A and 11 shows that the absolute values of reflectance are different, but the trend in brightness remains the same. This is because CS and SCS are independent of wavelength. However, in reality, the trend in reflectance in Sentinel-2 images differs between band 8A and 11, because there are other factors that reduce the reflectance besides shadows.

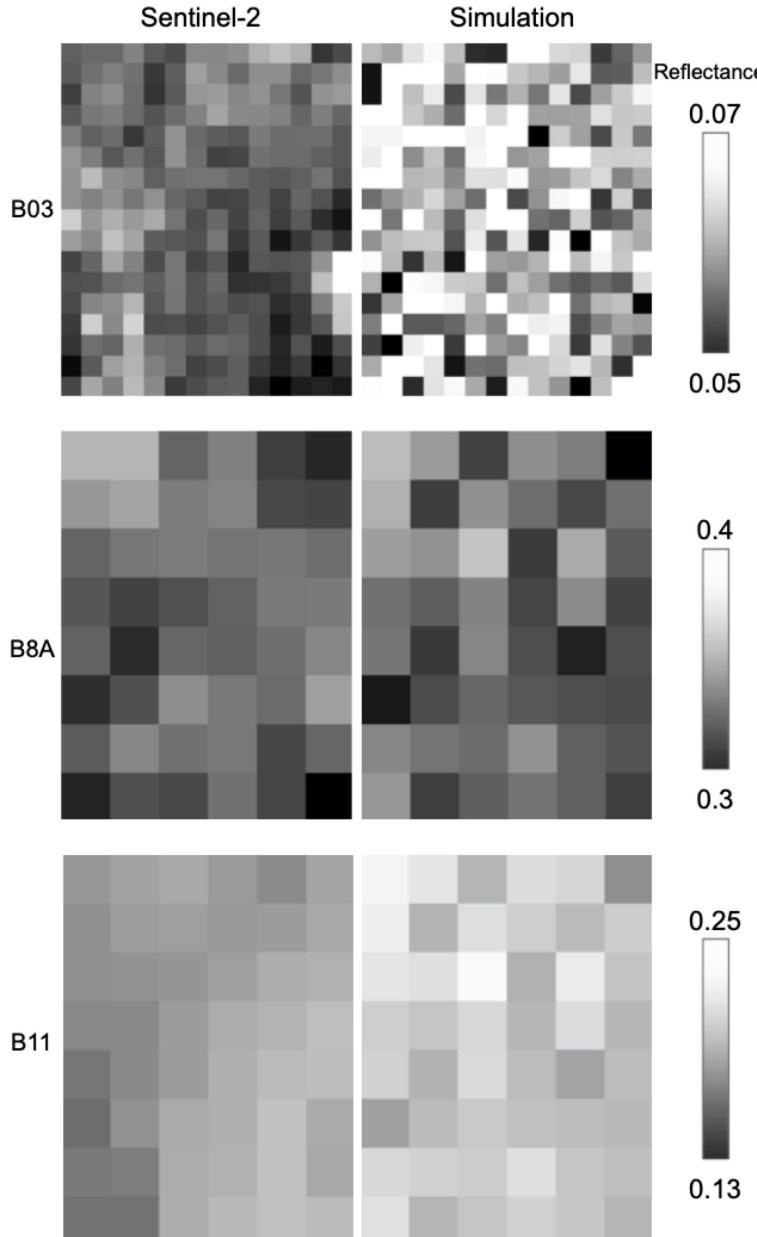

**Figure 8.** Comparison between simulated and acquired images of the forest. As an example, images in bands 3, 8A, and 11 simulated using *Quercus ilex* spectral reflectance are shown. The images in bands 3 and 11 are brighter than the Sentinel-2 images, indicating that these bands simulate higher reflectance than Sentinel-2.

Figure 9 shows the spectral reflectance profiles for the simulation results, Sentinel-2 TOA, and Sentinel-2 BOA. In this simulation, nine leaves spectral reflectances were used, as mentioned in Section 2.3.3. Not all of the pixels shown in Figure 8 were included in the creation of the spectral reflectance profile because there were areas of exposed soil in the target forest. The number of pixels used to create that profile was 185 pixels for bands 2, 3, 4, and 8 with a spatial resolution of 10 m, and 39 pixels were used for the other bands. When comparing the Sentinel-2 TOA to the BOA, the atmospheric correction appears to be correct. It is clear from Figure 9 that the simulation results are all overestimated for the Sentinel-2 BOA reflectance in bands 11 and 12 of SWIR. In the VIS and NIR bands, depending on the type of leaf, there is a case of over or underestimation. It is also found that the trends in spectral reflectance are different, even for the same species. Table 3 shows RMSE of the simulated reflectance to the Sentinel-2 BOA reflectance per band and its mean and Standard Deviation (StD). Simulations

using *Betula papyrfera* had the smallest RMSE. In particular, bands 2, 3, and 4 show very small RMSE of 0.06, 0.06, and 0.05, respectively. When *Quercus virginana* or *Quercus lobata* were used for simulation, the RMSE was twice that of *Betula papyrfera* at 0.040, but the StD was almost the same at 0.016. On the other hand, *Quercus douglasii* showed the largest RMSE; however, the decreasing trend in simulated reflectance using this leaf from NIR to SWIR bands is more similar than in other leaves.

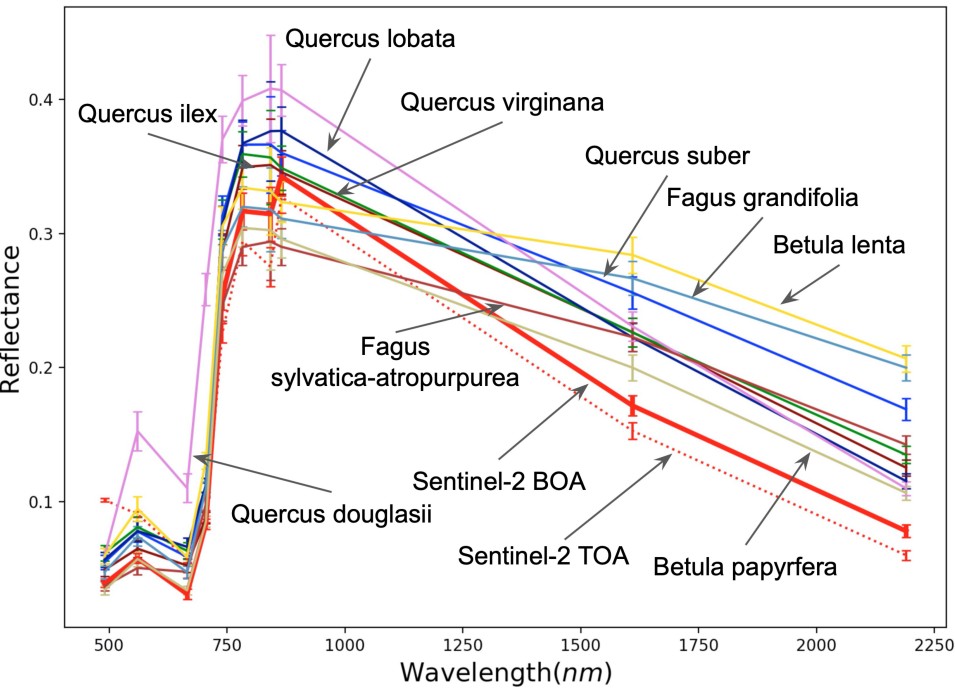

**Figure 9.** Spectral reflectance profiles for the simulation using the leaf spectral reflectance data as mentioned in Section 2.3.3, Sentinel-2 TOA and Sentinel-2 BOA. Mean and standard error of simulations is shown. The simulation results are all overestimated to Sentinel-2 BOA reflectance in SWIR (band 11 and 12).

**Table 3.** Root Mean Square Error (RMSE) of the simulated reflectance to the Sentinel-2 BOA reflectance per band and its mean and Standard Deviation (StD). Simulations using *Betula papyrfera* had the smallest RMSE (Mean RMSE is 0.20) and *Quercus douglasii* had the largest one (Mean RMSE is 0.84).

| Leaf | B02 | B03 | B04 | B05 | B06 | B07 | B08 | B8A | B11 | B12 | Mean | StD |
|---|---|---|---|---|---|---|---|---|---|---|---|---|
| Quercus ilex | 0.012 | 0.010 | 0.022 | 0.007 | 0.035 | 0.037 | 0.050 | 0.017 | 0.056 | 0.048 | 0.029 | 0.018 |
| Quercus virginana | 0.024 | 0.025 | 0.033 | 0.029 | 0.058 | 0.046 | 0.054 | 0.018 | 0.055 | 0.057 | 0.040 | 0.016 |
| Quercus suber | 0.017 | 0.021 | 0.028 | 0.026 | 0.061 | 0.052 | 0.063 | 0.024 | 0.085 | 0.091 | 0.047 | 0.027 |
| Quercus lobata | 0.019 | 0.022 | 0.036 | 0.029 | 0.054 | 0.053 | 0.072 | 0.038 | 0.052 | 0.038 | 0.041 | 0.016 |
| Quercus douglasii | 0.022 | 0.096 | 0.080 | 0.175 | 0.118 | 0.084 | 0.101 | 0.067 | 0.060 | 0.032 | 0.084 | 0.044 |
| Fagus sylvatica-atropurpurea | 0.005 | 0.009 | 0.017 | 0.016 | 0.013 | 0.031 | 0.036 | 0.055 | 0.052 | 0.065 | 0.030 | 0.021 |
| Fagus grandifolia | 0.010 | 0.018 | 0.017 | 0.027 | 0.037 | 0.016 | 0.032 | 0.035 | 0.096 | 0.122 | 0.041 | 0.037 |
| Betula papyrfera | 0.006 | 0.006 | 0.005 | 0.005 | 0.020 | 0.020 | 0.033 | 0.049 | 0.030 | 0.029 | 0.020 | 0.015 |
| Betula lenta | 0.022 | 0.038 | 0.028 | 0.048 | 0.053 | 0.024 | 0.037 | 0.025 | 0.113 | 0.129 | 0.052 | 0.038 |

## 3.4. Effect of Voxel Size on a Simulated Reflectance

The effect of the voxel size on the simulation was investigated. The voxel sizes used were 50 cm, 100 cm, and 200 cm. Firstly, the reflectance of each band was simulated from the voxel model for each size. Next, the average reflectance was calculated using the pixels that are mentioned in Section 3.3. Finally, the relative reflectance was then calculated by dividing that reflectance by the reflectance simulated from 20 cm voxel size. Table 4 shows that the relative reflectance increases as the voxel size increases from 20 cm to 200 cm, which is less than 1.05 at 50 cm, 1.1 at 100 cm, and 1.2 at 200 cm. This trend was the same for all of the bands. This result shows that, in this study, even when the voxel size was changed from 20 cm to 200 cm, the reflectance changed only 1.2 times.

**Table 4.** Relative reflectance in each band at voxel sizes of 50, 100, and 200 cm. After calculating the average reflectance across the forest from each voxel size, reflectance was derived by dividing these with simulated reflectance while using 20 cm voxel size. In this forest, the reflectance simulated from the 20- and 200-cm voxel models only varied about 1.2 times.

| Band | 50 cm | 100 cm | 200 cm |
|------|-------|--------|--------|
| 2 | 1.02 | 1.10 | 1.17 |
| 3 | 1.02 | 1.09 | 1.17 |
| 4 | 1.04 | 1.10 | 1.19 |
| 5 | 1.04 | 1.11 | 1.18 |
| 6 | 1.04 | 1.10 | 1.18 |
| 7 | 1.03 | 1.10 | 1.18 |
| 8 | 1.04 | 1.10 | 1.17 |
| 8A | 1.03 | 1.10 | 1.18 |
| 11 | 1.02 | 1.10 | 1.17 |
| 12 | 1.04 | 1.10 | 1.19 |

*3.5. Discussion*

When a point cloud is missing, it affects the results of the CS and SCS calculations. It is pointed out that if the point cloud is incomplete, points that belong to the shade component are identified as the sunlit component [27]. In UAV-SfM, the main causes of point cloud missing are shadows and occlusions [40]. In this regard, as mentioned in Section 2.3.1, this study was cloudy at the time of photography, so it can be inferred that there are no significant shadows in those images. Leaf shaking also affects the SfM algorithm and causes noise in the point clouds [40]; however, we did not perform noise reduction for the point cloud. Because there was scant wind, the swaying, if any, was likely to be a few centimeters in size. Moreover, the minimum voxel size in this study was 20 cm, so the impact was considered to be small. In the case of outliers, the effect was expected to increase as the voxel size was increased. However, the effect of outliers was also considered to be small because the relative reflectance changed very small, as shown in Table 4. In addition, because the georeferencing was performed based on the GPS log data installed in the UAV, the absolute error has not been calculated. If the error is larger than the voxel size, then it may affect the simulation results. Although this study collected images from a single camera angle, it has been reported that combining multiple angles provides a more comprehensive point cloud acquisition [41].

The spatial variability of the CS and SCS was clearly different, as shown in Figure 7. When compared to CS, SCS has a smaller variation with respect to the observation angle. This is because the sun position is used in the calculation of the CS, but not in the calculation of the SCS. These differences in spatial variability are related to structural parameters, such as forest density and roughness. Therefore, a multi-angle satellite would be expected to be able to estimate the variation of these shadows. CS and SCS are the shielding ratios for direct and diffuse solar irradiance, respectively, as shown in Equation (5). Therefore, the radiance is affected by both CS and SCS in the short-wavelength bands. On the other hand, the radiance is affected by only CS in the long-wavelength bands, because diffuse solar irradiance is reduced. A Second-generation GLobal Imager (SGLI) product provides the shadow index (SDI) that indicates the shadow content within a pixel as a new vegetation characteristic [42]. Because SDI uses the SWIR band, it means that value is CS. Because SGLI is also capable of multi-angle observations, it is expected to estimate structural parameters from the SDI obtained at each observation angle. On the other hand, since multi-directional observation is not possible with Sentinel-2 and Landsat, one can only extract the parameters of the forest structure by using the shadow changes due to changes in solar altitude. However, with multi-temporal images, reflectance changes not only due to shadows, but also due to changes in phenology [43,44]. Therefore, the challenge for the future is to understand the impact of shadows and phenology on the annual change in reflectance rates.

The best RMSE was 2 ± 1.5% when the reflectance was simulated using *Betula papyrfera*, as shown in Table 3. This result is comparable to the accuracy of Rengarajan and Schott. [45] when they used the Digital Image and Remote Sensing Image Generation tool to simulate the NIR band reflectance of Landsat-8 surface reflectance product. Usually, at short wavelengths, the simulation error is expected to be larger due to the influence of path-radiance. On the other hand, at long wavelengths, the simulation error is smaller because of its negligible impact. However, the present results are the opposite, as shown in Figure 9 and Table 3. That is, the reflectance simulated in bands 11 and 12 were overestimated. The simulation method proposed in this study has four wavelength-dependent elements. They are SMARTS, Sen2Cor, the spectral reflectance of the leaves, and the water content of that one. The atmospheric conditions at the date of the Sentinel-2 image used were fine, so the parameters used for SMARTS and Sen2Cor were also standard value. In addition, the similarity of the spectral reflectance shapes of Sentinel-2TOA and BOA shown in Figure 9 suggests the atmospheric effect is small. Gueymard. [38] has calculated the absolute spectral difference between SMARTS and MODTRAN4 irradiance predictions under air mass 1.5 and standard atmospheric conditions. The result showed that the absolute spectral differences were within 5% for all wavelengths. In addition, Li et al. [46] validated the Sentinel-2BOA reflectance generated using Sen2Cor using the solar spectrum-vector (6SV) code. The results showed that the BOA reflectance was overestimated over 6SV in all bands. The reflectances of nine different leaves types were used in the simulations in this study, but all of them were overestimated in bands 11 and 12. Therefore, the spectral reflectance used was also not considered to be the cause of the overestimation. Seelig et al. [47] reported a large decrease in reflectance in SWIR as compared to VIS and NIR as the relative water content of leaves increased. In the leaves they studied, even a 20% difference in relative water content changed the reflectance by about 0.05 in the SWIR band. Although there was no precipitation before or after the date acquired by Sentinel-2, the water content of the leaves used in the simulation was lower than that of the target forest, which may be the reason for the overestimation in SWIR bands.

As the voxel size increased from 20 cm to 200 cm, the relative average reflectance increased by the same ratio in all bands (Table 4). The reason for that is the increased voxel size reduces the roughness of the three-dimensional (3D) model and the effect of CS and CSC was decreased. This finding was expected to estimate errors, even when a bigger voxel size was used for reducing the computational cost. However, this trend may change, depending on the type of forest. In the future, we need to create forests with different densities and roughnesses to further investigate the impact of voxel size on the simulation.

The main model limitation of this simulation was that all leaf orientations were assumed to be perpendicular to the XY plane, as described in Section 2.4.2. This assumption is expected to lead to a reduction in simulation accuracy in the case of rougher forests, such as coniferous forests. The distribution of leaf inclination angles also has a direct effect on the fraction of solar radiation that is shielded [48]. Therefore, it is necessary to use the point cloud in the voxel in order to estimate the orientation of the leaves. Furthermore, all of the voxels are assumed to be leaves. In reality, however, the voxels include branches, forest floor, and other objects with different reflectance. To solve this problem, voxels need to be classified according to each object and each reflectance be assigned.

## 4. Conclusions

In this study, voxel models with CS and SCS as attributes were developed and their spatial variations were investigated. In addition, a reflectance simulation method was developed and compared with the Sentinel-2 BOA reflectance. A voxel model was created based on the point clouds that were obtained by UAV-SfM in the target forest. In the investigation of the spatial variability, we fixed the position of the sun and changed the view zenith angle from $-30°$ to $30°$. The result showed that the spatial variability of the CS was greater than that of the SCS. The best RMSE for the simulated reflectance to Sentinel-2 BOA reflectance was 2 ± 1.5%. The spectral reflectance of nine different leaves was used, but all of the simulations resulted in an overestimate in the SWIR

bands. Atmospheric corrections were also good, so the cause was considered to be leaf water content. The effect of the voxel size on the simulation was investigated. The voxel sizes used were 50 cm, 100 cm, and 200 cm. The results showed that the relative average reflectance increases at the same rate in each band. This result contributes to the error estimation when the voxel size is increased to reduce the computational cost. The method developed in this study is useful for studying the BRDF and estimation of physical quantities using shadow proportions. The closest Sentinel-2 image to the date of the UAV observations were used for comparison. Future work will include simulating reflectance by considering the inclination of leaves and simulating multi-temporal satellite images in order to investigate the effects of phenology and shadow on reflectance variation.

**Author Contributions:** Conceptualization, Takumi Fujiwara and Wataru Takeuchi; Methodology, Takumi Fujiwara; Software, Takumi Fujiwara; Validation, Takumi Fujiwara; Formal Analysis, Takumi Fujiwara; Resources, Takumi Fujiwara; Data Curation, Takumi Fujiwara; Writing-Original Draft Preparation, Takumi Fujiwara; Writing-Review & Editing, Wataru Takeuchi; Visualization, Takumi Fujiwara; Supervision, Wataru Takeuchi; Project Administration, Wataru Takeuchi. All authors have read and agreed to the published version of the manuscript.

**Funding:** This research received no external funding.

**Conflicts of Interest:** The authors declare no conflict of interest.

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
