# Peer review of "Simulation of Sentinel-2 Bottom of Atmosphere Reflectance Using Shadow Parameters on a Deciduous Forest in Thailand"

_ijgi, doi:10.3390/ijgi9100582_

Round 1

Reviewer 1 Report

The objectives of this paper were: 1) to investigate the spatial variations of CS and SCS, 2) to compare the satellite-observed and simulated reflectance, 3) to investigate the effect of voxel size on simulation result. It used the Bottom Of Atmosphere (BOA) reflectance image from Sentinel-2. This study can contribute for understanding BRDF in forest. The paper is based on the knowledge available in the literature. The English structure of the manuscript could be improved to be clear for the readers.

Some corrections and suggestions:

L.6 – and it –-- and the reflectance

L.11- nine leaf  --- nine leaves

L.19- The satellite data obtained from optical satellite imagery  ---- The satellite data obtained from optical sensors

L.29 - SI correlated higher than --- ?

L.39- satellite data.  ---  satellite sensor data.

L.43- simulators input vegetation parameters  ----  simulators use vegetation parameters

L.81- position, and voxel model  ---  position, voxel model

L.82 - were input to simulate ---  were used to simulate

L.83 - was assumed leaf voxel.  ---  was assumed as leaf voxel.

L.85 - Finally, the effect of voxel size on simulation result was confirmed. ---  Finally, the effect of voxel size on simulation result was investigated.

L.89- The target extent was 170m_130m and the elevation was about 230 m ---  The target extent is 170m_130m and the elevation is about 230 m

L.94 - The site was dense forest and soil was exposed on some area. ---  The site is dense forest and soil is exposed on some area.

L.98 - The UAV used was  ---  The UAV used is

L.103- acquired on 29 the September --- acquired on 29th September

L.106- target band is 2, 3, 4,  5, 6, 7, 8, 8A, 11 and 12.  --- the bands are 2, 3, 4,  5, 6, 7, 8, 8A, 11 and 12. --- I suggest to write the wavelength intervals of these bands.

Table 1 --- Patameter --- Parameter

                 Grounf ---  Ground

L.119- Nine leaf data  ---  Nine leaves data

Figure 3 -  nine leaf  --- nine leaves

L.127- voxels were used to calculate CS and SCS was calculated.  ----  voxels were used to calculate CS and SCS.

L.128- in the voxel is decrease.  ---  in the voxel decreases.

L.140- azimuth angel ---  azimuth angle

Figure 7- (red dotted line)  --- (red line) 

L.207 - nine leaf  ---  nine leaves

L.207- as mentioned in Sanction 2.3.3  ---   as mentioned in Section 2.3.3

L.225 - Because,  it is pointed out that  ---  It is pointed out that

L.229- so the it can be inferred ---  so it can be inferred

L.230- resulting it makes noise point clouds  ---  resulting noise in the point clouds

L.261- was an overestimate.  ---  were overestimated.

L.271- is overestimated in bands 11 and 12 than BOT reflectance.--- ?

L.271- The reflectance of nine different leaf types was used ---  The reflectances of nine different leaves types were used

L.300- in the SWIR  ---  in the SWIR bands.

L.301- nine different leaf reflectances  ---  nine different leaves reflectances

Reviewer 2 Report

Review: Simulation of Sentinel-2 Bottom of Atmosphere Reflectance Using a Shadow parameters on a deciduous forest in Thailand

This paper studied the relationship between shadow and radiance and it provides basic foundation for understanding BRDF in forest. I have the following specific comments:

Abstract,

Line 2-3, “physical quantities, it is not clear what does it refer to? Physical quantities of forest? Does it refer to biomass calculation?

Line 6, please clarify “it”,

Line 6-8, This sentence includes two different ideas that are not evidently related one to the other. So please rewrite this sentence. The connection between the problem of forest BRDF and Sentinel-2 bottom of atmosphere reflectance is not evident, please provide more background for it.

Line 11, “…Spectral reflectance data for nine leaf were used in the simulations”, but Sentinel-2 image does not reflectance at the leaf level,

Introduction

Line 31-33, based on the explanation that the authors provided in this sentence, it is not very clear why the semi-empirical model can not be applied to high-resolution satellite images. Was it because the high-spatial resolution images do not have enough temporal resolution to meet the requirement of the minimum of 7 valid 16-day reflectance data? How about a build between MODIS BRDF and high-resolution data (see Li et al 2017-Improving BRDF normalization for Landsat data using statistical relationships between MODIS BRDF shape and vegetation structure in the Australian continent, RSE, 196, 275-296).

Line 36, please specify the “various parameters”

Line 69, compare instead of “comparing of”

Methodology

Line 91, “…no observations have been made on this tree species”. Please specify 

Table 1, Parameter, Ground elevation instead of “Grounf elevation”

Line 120, Here the data of the nine leaves for Quercus, Fagus, and Betula are to represent the dominant tree species Deciduous dipterocarp? This information is missing in the study area,

Figure 3, if the main leaf genera are three types Quercus, Fagus, and Betula, why use a total of nine leaf species spectral reflectance data?

Line 127, remove “was calculated”.

Line 128, is decreased.

Line 159, “…with a pixel size was same with voxel size…” remove was.

Line 162, change inputted to input

Figure 8, I found this figure difficult to interpret, too crowded,

Reviewer 3 Report

This article aims to create a voxel model with Cast Shadow (CS) and Self Cast Shadow (SCS) as attributes based on point cloud, and investigate their spatial variations, compare the satellite observed and simulated reflectance based on the voxel model, and evaluate the influence of voxel size on simulation reflectance. This work is useful for studying the BRDF of forest and estimation of physical quantities using shadow proportion. The authors of this paper have good experience in the voxel model and forest canopy simulation, which makes this work desirable.

This article considers the three-dimensional structure, obtains the point cloud to establish the voxel model to study the CS, and SCS is a big innovation point. In general, this idea is reasonable and can be used as a reference for other research on the relationship between shadows and satellite data. However, this article still has some problems with the research process and writing. I suggest a Major revision on both writing and science parts.

My main concerns are as follows, and I recommend that the issues to be explained in detail before the paper can be published.

  1. The Abstract lacks a direct description of the research process, and I suggest to give a concise summarization before the Results to make the content more complete.
  2. I recommend adding some related keywords to facilitate readers to search,such as “shadow”.
  3. The content of Section 3.1 may be more suitable to be placed in methodology.
  4. This study did not perform noise reduction for the point cloud, whether this will cause too much error in the result. In addition, it cannot prove that the results are universal. Too many uncertain factors will cause the results to be unconvincing. You may give a reasonable discussion about this.
  5. Since you are simulating the signal of an existing sensor, I expect to see the comparison between simulated and acquired images of the forest. Only comparing the reflectance curves may not be enough.
  6. You may add more description about your results. Currently, the paragraphs are too short to give enough information.
  7. There are obvious grammatical errors in the article, please check and modify them carefully. E.g. 1) Misuse of the third-person singular number:in line 78, "CS and SCS mean"; in Line 108, “VIS, NIR, and SWIR was simulated”; in ine 138, “the method project”. 2) The sentence does not make sense: in line 80-81, “in Section 2.4 Thirdly, after deciding viewing position, and voxel model was converted to image”. 3) Punctuation miscues:in line 80,there is lack of a comma before“Thirdly”.
  8. There are also formatting errors in the article, please check and modify them carefully. E.g. 1) In line 79, "direct-" needs to delete "-". 2) In Line 107, "visible (VIS), near infrared (NIR) and short wavelength infrared (SWIR)", the letter in full spelling corresponding to the abbreviation should be capitalized.
  9. In Table 1, "Grounf" seems to be spelled incorrectly. The author needs to pay more attention to the details before resubmitting this paper.

Round 2

Reviewer 1 Report

The objectives of this paper were: 1) to investigate the spatial variations of CS and SCS, 2) to compare the satellite-observed and simulated reflectance, 3) to investigate the effect of voxel size on simulation result. It used the Bottom Of Atmosphere (BOA) reflectance image from Sentinel-2. This study can contribute for understanding the BRDF of forest. The paper is based on the knowledge available in the literature.

Some correction:

L.124 - In this study, the bands are 2, 3,  4, 5, 6, 7, 8, 8A, 11 and 12. The wavelength intervals of each band are 66 nm, 36 nm, 31 nm, 106 nm,  16 nm, 15 nm, 20 nm, 22 nm, 94 nm, and 185 nm. Thus, VISible (VIS), Near InfraRed (NIR) and Short  Wavelength InfraRed (SWIR) were simulated target wavelength. --- In this study, the bands are 2, 3,  4, 5, 6, 7, 8, 8A, 11 and 12. The central wavelength of each band are 490 nm, 560 nm, 665 nm, 705 nm,  740 nm, 783 nm, 842 nm, 865 nm, 1610 nm, and 2190 nm, respectively. Thus, VISible (VIS), Near InfraRed (NIR) and Short  Wavelength InfraRed (SWIR) were the simulated target wavelength.

Reviewer 3 Report

This reversion solved all my concerns.  You may double check the language and other details before painted. 
